# Option talk and risk communication with people with limited health literacy: A qualitative focus group study with key stakeholders

Romy Richter[1,*,☯], Esther Giroldi[1,2], Maarten Jonkmans[1,‡], Juliette Linskens[1,‡], Jany Rademakers[1,3,☯], Trudy van der Weijden[1,☯], Jesse Jansen[1,☯]

**1** Department of Family Medicine, Care and Public Health Research Institute CAPHRI, Maastricht, Netherlands, **2** Department of Educational Development and Research, Faculty of Health Medicine and Life Sciences (FHML), School of Health Professions Education (SHE), Maastricht University, Maastricht, Limburg, Netherlands, **3** Nivel, Netherlands Institute for Health Services Research, Utrecht, Netherlands

☯ These authors contributed equally to this work.
‡ MJ and JL also contributed equally to this work.
* romy.richter@maastrichtuniversity.nl

## Abstract

### Purpose

To explore stakeholders' views on acceptable and feasible strategies for discussion of treatment options and risk communication with people with limited health literacy (LHL in the context of shared decision-making (SDM)).

### Methods

This qualitative descriptive study used purposive sampling to conduct focus groups with stakeholders including experts in health literacy, SDM, or risk communication (RC); experienced General Practitioners (GPs); and individuals with LHL. Each session included a brief presentation defining SDM, option talk, and RC, followed by an introduction to various RC strategies and decision aids to facilitate discussion. Verbatim transcripts were analysed by two independent researchers using inductive and deductive content analysis.

### Results

Five focus groups were conducted, involving experts (two FGs, n = 5 and n = 6), GPs (one FG, n = 8) and people with LHL (two FGs, 2x n = 3). Experts and GPs emphasised the need to tailor communication to the patient's context and noted challenges in identifying patients with LHL. All participants highlighted the importance of using illustrations of treatment options (e.g., knee injections) to support the discussion of options. Views on the level of detail required for RC varied, with some GPs questioning whether RC was understandable to people with LHL. Most people with LHL preferred RC in natural frequencies and icon arrays but noted that RC can fan fear. GPs

**Data availability statement:** All relevant data are within the manuscript and its Supporting Information files. The submission contains all raw data required to replicate the results of our study.

**Funding:** This work was funded by THE NETHERLANDS ORGANISATION FOR HEALTH RESEARCH AND DEVELOPMENT (ZonMw) [project number 10060011910007, 2021]. https://www.zonmw.nl/en. The funders played no role in study design, data collection and analysis, decision to publish, or preparation of the manuscript.

**Competing interests:** The authors have declared that no competing interests exist.

found contextualisation (e.g., comparing the probability of a treatment outcome with the probability of a car accident) a helpful strategy, but patients found it confusing. Decision aids were seen as supportive for RC. Overall, people with LHL preferred their doctor to discuss options face-face with them, using a layered step-by-step manner, adding details on RC as preferred by patients.

## Conclusion

SDM for people with LHL benefits from a tailored, layered approach with visual aids. These strategies are potentially useful for all patients, but further research is needed to confirm this.

---

## Introduction

For patients with limited health literacy (LHL), making healthcare decisions is challenging. Shared decision-making (SDM) has increasingly become a gold standard for healthcare provider-patient communication. SDM is a process in which the healthcare provider and patient discuss which medical management plan is best for the patient. This involves considering all relevant options (option talk) with their benefits and harms and the patient's preferences and context [1]. Communicating the benefits and harms of screening or treatment options is called risk communication (RC). It is an integral part of patient education, as everyone has the right to comprehensive, evidence-based information about the benefits and harms of health interventions [2]. Campaigns like "Ask three questions" [3] and "Begin een goed gesprek" [start a good conversation] promote SDM awareness [4]. SDM enhances patient participation, knowledge, informed choice, and reduces decisional conflict [5]. Deciding together is a step towards involving all patients in decision-making and tailoring appropriate care.

Health literacy is crucial for understanding health information and making informed health decisions. It entails people's knowledge, motivation, and competencies to access, understand, appraise, and apply health information to make judgments and decisions about their health [6]. Across Europe, 27% to 48% of individuals have LHL [7], with one in three Dutch affected [8]. People with LHL often participate less in health decision-making, partly due to lower perceived self-efficacy [9] and lower patient activation [10]. They also have more difficulty understanding medical information, conditions, and care and struggle to use medication correctly [9]. Chronic disease management in this group is often poor, and they are more likely to engage in riskier health choices, such as smoking and poor medication adherence [11].

RC is a key challenge in SDM [12] and is typically communicated through verbal, numerical or visual formats [13–15]. Verbal RC alone, e.g., "your risk is low", leads to ambiguity in interpretation [16], while numerical RC using natural frequencies or percentages, is preferred over verbal RC only [13]. Visual RC strategies like icon arrays, bar charts or risk ladders further enhance understanding [17,18]. Risk information can be framed positively or negatively; for example, an 80% chance of survival versus a 20% chance of dying influences risk perception and decision-making [15].

Clear presentation of medical options, known as option talk and RC, is essential for informed decision-making [15]. Patient decision aids (PtDAs) can be used to support this process by explicitly presenting probabilities of benefits and harms of, e.g., treatment options [19]. Strong evidence shows that using PtDAs results in improved knowledge, more realistic expectations, and increased patient involvement in decision-making. A meta-analysis found PtDAs particularly beneficial for people with LHL, offering greater advantages than for those with higher literacy, education, or socioeconomic status [5]. Research also confirms that visualisation of options and probabilities enhances understanding and decision-making, especially for people with LHL [17,18].

Overall, research on RC for people with LHL, particularly qualitative studies that provide insight into stakeholders' perspectives in clinical practice, is scarce. This qualitative study builds on our previous work on RC strategies and analyses of PtDAs [12,20] within a project to improve option talk and RC for people with LHL. To gain an understanding of acceptable and feasible strategies to support SDM in practice, we aim to explore views and preferences regarding the discussion of treatment options (option talk) and RC among (a) experts in SDM, health literacy and RC, (b) experienced GPs, and (c) people with LHL.

## Materials and methods

### Study design

This qualitative descriptive study with focus groups (FG) took the ontological orientation of relativism, which holds the view that multiple subjective realities exist. Consequently, we chose the epistemological orientation of subjectivism, which acknowledges multiple interpretations of reality. Since participants' RC strategies are subjectively developed in relation to the clinical situation [21]. The focus group methodology was used as it combines interviewing several people and group interaction so that participants can immediately react to each other's perspectives on the complex topic of RC. This reveals experiences with strategies, challenges, and contradicting aspects, fostering reflection and collaborative construction to gain a deeper understanding [22–24]. The consolidated criteria for reporting qualitative research (COREQ) have been used to guide reporting of the research [25].

### Conceptual framework

Discussing options (option talk) and RC is part of SDM [1]. RC is defined as: "*The open, two-way exchange of information and opinion about risk [benefits and harms], leading to a better understanding of the risk [benefits and harms] in question, and promoting better (clinical) decisions about management*" [26]. A deductive conceptual framework was developed in our previous work to display a theoretical foundation and establish current knowledge regarding RC strategies [12] (Figs 1 and 2). This framework guided the data collection and analysis process.

### Participant selection

We used purposive sampling. Eleven experts and 30 GPs were contacted by email through the research team and their network from 01/10/2021 to 31/01/2022. To recruit GPs, names and contact details of potential participants were gathered in consultation with project advisors and with a working group on diversity and international health from the Dutch College of GPs (DCGP). All the experts and ten GPs responded positively to participating. Six people with LHL were recruited by The Dutch Centre of Expertise on Health Disparities (PHAROS) between February and May 2022. All participants had to be proficient in the Dutch language. Inclusion criteria can be found in Table 1.

### Data collection

**Interview guides.** The researchers designed the semi-structured interview guides collaboratively based on the conceptual framework and findings of previous work on RC [12,20,28]. The interview guides were divided into two

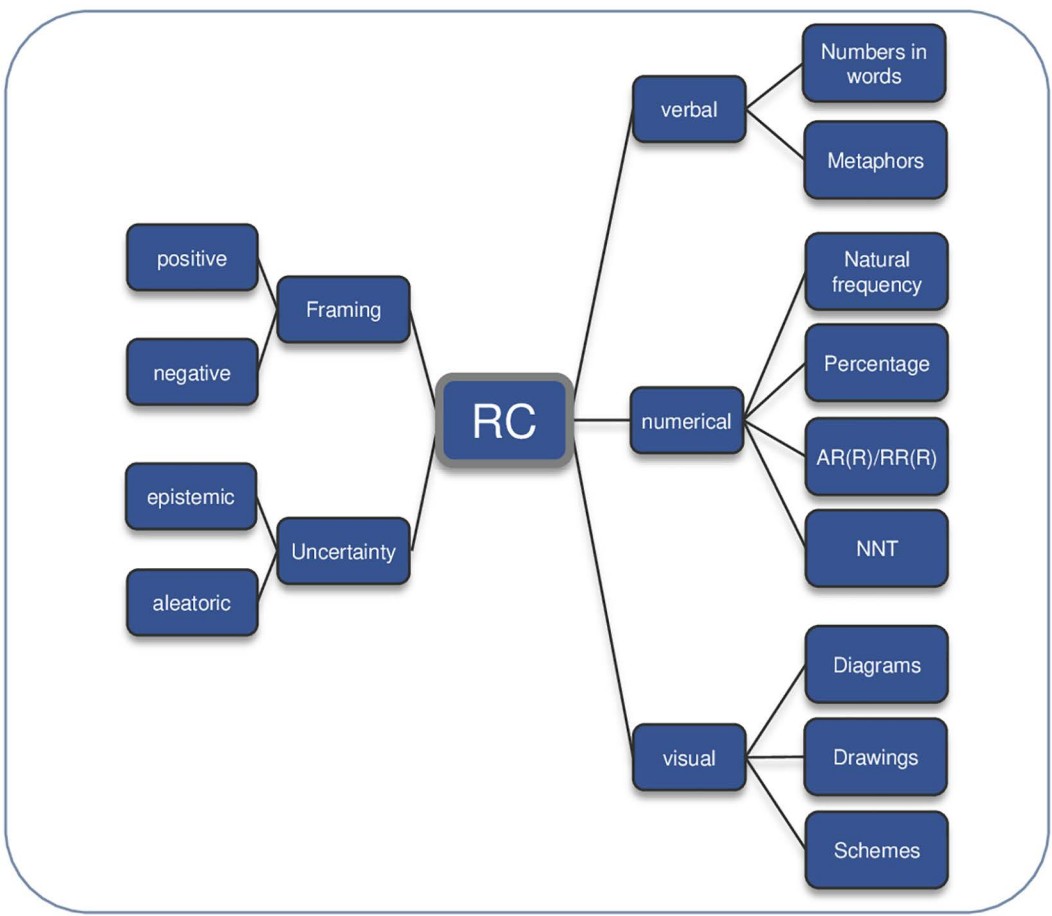

**Fig 1. Framework RC strategies [12]** AR(R) = Absolute Risk (Reduction), RR(R) = Relative Risk (Reduction), NNT = Number Needed to Treat.

main topics: a) Participants' perspectives on and experiences with option talk and RC with patients with LHL/with their physician, and b) Participants' perspectives on using Dutch PtDAs and RC strategies for people with LHL. The main questions were followed by probing questions to better understand the participants' answers. For the FG with experts and GPs, the interview guide consisted of an introduction, a briefing, the two discussion topics and a closing. The briefing had the same structure as the background information. Background information consisted of a summary of the project and findings (SDM model and project phases, summary of RC strategies based on the literature review, insights into PtDA analysis and examples). Providing this information aimed to focus the participants on the topic of option talk and RC with patients with LHL. At the start of the FGs, we defined the key terms SDM, RC and health literacy to ensure the same understanding by the interviewer and participants. During the FGs with experts and GPs, three PtDA format examples about knee arthritis were presented to inspire discussion on those formats (S1 Appendix). Participants with LHL received no information prior to the FG to prevent information overload but were informed about the aim of enabling informed consent. For people with LHL, examples of a PtDA and verbal, numerical and visual RC strategies were provided during the interview to stimulate discussion (S2 Appendix). The interview guide for people with LHL was pilot tested by PHAROS with four people with LHL to test the structure and comprehensibility of questions and figures.

 **FG interviews.** Due to the COVID-19 pandemic, FGs with experts and GPs took place online via the video platform Zoom©, all but two of the GPs who were approached participated in the study. The FGs with experts and GPs were

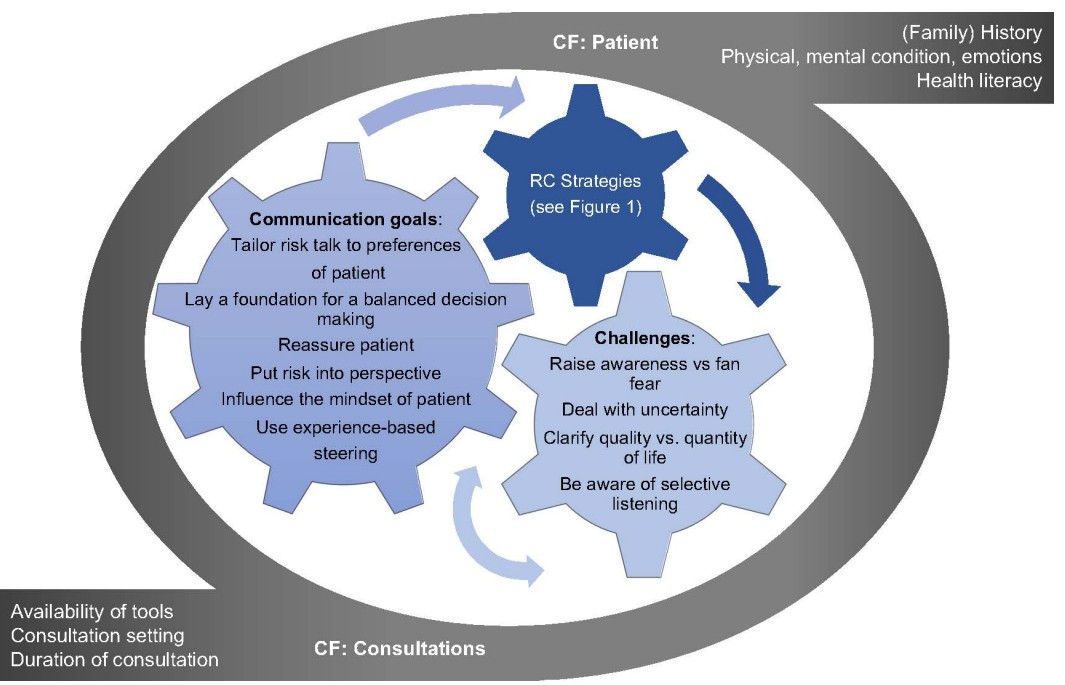

**Fig 2. Framework RC in clinical practice [12].**

**Table 1. Inclusion criteria.**

| Participant Group | Inclusion criteria |
|---|---|
| **Experts** | having a PhD in the field of SDM and/or in the field of health literacy and/ or RC and/ or are knowledgeable on PtDAs and their implementation for LHL-patients. |
| **General Practitioners** | GPs who regularly encounter patients with LHL. |
| **People with LHL** | people who are/were patient (> a single short time encounters) and met at least one of the following criteria: low income, low social support, living situation in disadvantaged district, low education or LHL.* |

*Established by PHAROS based on [27].

conducted in January and February 2022. The FG with people with LHL took place at PHAROS in May 2022. The FGs were conducted in Dutch and lasted one and a half hours, with a break between the two main topics. FGs were audio and/ or video recorded, and field notes were taken during the FGs with experts and GPs. We started with the two expert FGs, followed by the FG with the GPs and ended with the two FGs with people with LHL. A different interviewer was assigned to each participant group. The focus groups with individuals with LHL were conducted by an experienced interviewer from PHAROS. Focus groups with experts were led by co-author JR, a professor in health literacy, while those with GPs were conducted by co-author TvdW, a professor in shared decision-making. Both co-authors have extensive experience in conducting focus groups.

## Data analyses

All interviews were transcribed verbatim. We used deductive content analysis to gather RC communication strategies according to the categories described in the framework [12]. At the same time, an inductive approach was applied to

capture and categorise new aspects related to views on option talk and RC strategies without preconceived ideas. This included identifying challenges, dilemmas, and factors associated with content and context [29]. Pairs of researchers (RR and TvdW or RR and JL or MJ and TvdW) independently coded the transcripts based on categories of the RC framework and additionally coded further meaning units open-mindedly. The researchers reached a consensus on the final codes through discussion. Final codes and text fragments were subsequently captured in the software program NVIVO. Although data were collected through focus groups, the presentation of results also attends to individual perspectives expressed within the group discussions. This approach enables a more comprehensive and nuanced understanding of the data by highlighting diverse viewpoints while preserving the group context.

## Techniques to enhance trustworthiness

The richness of data was enhanced by including experts and GPs with diverse backgrounds. The representativeness of people with LHL was encouraged by assigning recruitment to an expertise Centre for health literacy. The semi-structured interview guides comprising broad questions left enough latitude for the participants to express their perspectives [30]. The research aim constantly drove the iterative data collection and analyses. Iteration enabled the subsequent FGs to be informed by refining prompting questions as needed, facilitating continuous meaning-making. The theoretical framework was critically verified, reflected upon, and expanded [12]. A member check for the FGs with experts and GPs enabled the participants to comment on the interview transcripts [21,31]. Two researchers independently coded transcripts. Analysis and interpretation were additionally discussed by the research team (investigator triangulation), aiming to avoid fundamental biases arising from the use of a single observer [30].

## Ethical clearance

The Medical Ethical Review Board (METC azM/UM) approved our study protocol (2021-2930). The experts and GPs were asked for verbal informed consent at the beginning of the focus group interview. People with LHL provided written informed consent prior to the focus group interview. Written informed consent from people with LHL was obtained and retained by the Dutch Centre of Expertise on Health Disparities (PHAROS).

## Results

### FGs and participant characteristics

We conducted five FGs. Table 2 provides further information on the FGs and participants. All participants attended the FGs on time, and the atmosphere was open and welcoming. There were no cancellations for the FGs with experts and individuals with limited health literacy (LHL), although two participants withdrew from the GP FG due to personal reasons. There were no negative power dynamics in any of the FGs.

**FG experts.** In both FGs with experts, there was a balanced representation of SDM and health literacy. Most of the input gathered in the first FG with experts was also mentioned in the second FG with experts. Only a few new themes emerged. Many of the experts were familiar with both each other and the moderator. In the expert FGs, more females than males participated.

**FGs GPs.** In the FG with GPs, most participants from the Dutch College of General Practitioners (DCGP) knew each other, and the moderator (TvdW) already knew three of the participants. Although we have no indication that this influenced the FG, we cannot completely exclude some influence. In the FG with GPs, one participant spoke a larger amount. There seemed to be much respect for this person's seniority and work experience. Hence, during the GP FGs, not all participants spoke equally. The moderator tried to stimulate participation by explicitly targeting the quieter participants.

**FGs people with LHL.** The gender ratio for the FGs with people with LHL was balanced. People with LHL obtained at least primary education except for one person. Five participants with LHL were Dutch; one had a migration background. One person had paid work.

**Table 2. Participant characteristics.**

| Experts* | | | |
|---|---|---|---|
| **Participant number** | **Gender** | **Expertise** | **FG number (n = 5)** |
| E1 | F | Public health and RC | 1 |
| E2 | F | Health care communication, palliative care, LHL | 1 |
| E3 | F | Medical psychology | 1 |
| E4 | M | Evidence-based medicine and SDM | 1 |
| E5 | F | Personalized medicine and quality of care | 1 |
| E6 | F | Health care communication (focus primary care) | 2 |
| E7 | F | Public and occupational health and methodology | 2 |
| E8 | F | Senior advisor foundation health literacy | 2 |
| E9 | F | Personalised care and support | 2 |
| E10 | F | General practice | 2 |
| E11 | F | Cognitive psychology | 2 |
| **General Practitioners** | | | |
| **Participant number** | **Gender** | **Years working experience** | **FG number (n = 5)** |
| G1 | M | 10 | 3 |
| G2 | F | 23 | 3 |
| G3 | M | 39 | 3 |
| G4 | M | 11 | 3 |
| G5 | F | 13 | 3 |
| G6 | F | 17 | 3 |
| G7 | F | 9 | 3 |
| G8 | F | 8 | 3 |
| **People with LHL*** | | | |
| **Participant number** | **Gender** | **Age Group** | **FG number (n = 5)** |
| P1 | M | 62 | 4 |
| P2 | F | 77 | 4 |
| P3 | M | 50 | 4 |
| P4 | M | 62 | 5 |
| P5 | F | 63 | 5 |
| P6 | F | 52 | 5 |

*Focus groups with experts and people with LHL were divided in two groups. Grey shading indicates these divisions.

## Main findings

The main categories that emerged from all FGs are:

- Challenges experienced in SDM with patients with LHL;

- Prioritising patient context and an iterative SDM process;

- Critical views on the feasibility of option talk and RC in daily practice;

- Preferences for specific RC strategies

- Critical views on the usability of PtDAs for people with LHL

The results are discussed below, illustrated with participants' quotes. Table 3 summarises recommendations on option talk, RC, and use of information material extracted based on the different stakeholders' views.

**Table 3. Recommendations for SDM in general and option talk, RC and use of information materials for patients with LHL (based on all FGs).**

| SDM in general |
| --- |
| Know/understand the context of the patient. (E, GP, P) |
| `Before discussing options, explore patients' knowledge of their context and beliefs, such as on diagnoses and disease (Step 0). (E) |
| Use the teach-back method. (E, P) |
| Provide space for patients to express their needs, beliefs, and preferences. (E, GP, P) |
| If possible, plan a second consultation to make a decision. (E, GP, P) |
| Offer the patient the opportunity to bring a relative or friend to the consultation. (P) |
| Offer the patient to take notes or to audiotape the consult. (P) |
| **Option talk** |
| Present all clinically relevant options to the patient. (P) |
| Be aware of information overload. Use a layered approach when giving information. (E, GP, P) |
| Use illustrations to explain the options. (E, GP, P) |
| **RC** |
| Ask patients how much RC detail they want. (E, GP, P) |
| Avoid the use of verbal RC only. (E, P) |
| Contextualisation seems to lead to confusion. (E, P) |
| Use natural frequencies instead of percentages. (E, GP, P) |
| Use numerical RC together with visual RC. (E, P) |
| Use icon arrays to visualise the probability of potential benefits and harms of treatment. (E, GP, P) |
| **Use of information material** |
| Consider digital literacy of patient, also offer printouts. (P) |
| Use less text and a clear layout. (P) |
| Use large fonts. (P) |
| Review the information with the patient. (E, GP, P) |

E = experts, GP = General Practitioner, P = patient: indicates the participant's group of which the recommendation is mainly based on.

**Challenges experienced in SDM with patients with LHL.** According to several experts and GPs, identifying patients with LHL is challenging due to the diversity in LHL characteristics. Subjective LHL assessment is usually based on patient records and background information from previous consultations. Some experts emphasised that the doctor needs to be aware that the knowledge gap between doctor and patient is not only about medical knowledge but also about differences in mental models. These beliefs are related to factors like patient context and emotions, social setting, and past healthcare experiences. Experts emphasised that communication should align with the patient's beliefs rather than the doctor's beliefs. They stressed making SDM understandable, not just for patients with LHL but for all patients, since a comprehensible consultation is desirable for all patients and health care providers.

*So, you have to bring people along and make sure they understand the context, and only then can you give information. [...] The expert sends it from his/her mental model, and then it arrives in the patient's mental model, and if there is space in between, it doesn't arrive. You have to map that out to develop a good relationship.* [E1]

As soon as you start with your medical story immediately, you leave the patient's domain. You should start within the patient's domain. *[E10]*

Several experts and GPs recommended applying checks during consultations to ensure patient understanding, for example, through the teach-back method. They suggested making the teach-back method questions specific and demarcated, for example, asking, 'How would you describe the two options to your husband?' instead of a general testing question such as, *'How would you summarise what I just explained about the options?'*

*Another important addition to this [teach-back method] is that you can do it in between, and you have to make it very specific. So not 'do you want to summarise what I just told you', but more 'about this treatment, what does it entail?'. So, checking very specifically. It's impossible for people to repeat everything. [E5]*

Likewise, patients with LHL regarded the teach-back method as a helpful tool. One participant with LHL suggested hanging posters or a screen in the waiting room to inform about the teach-back method. They discussed the importance of patients daring to interrupt the doctor if they needed help understanding something or if the doctor was continuing too fast. They mentioned that they were too anxious to do this in earlier years but are now taking a more active role in their decision-making. People with LHL disliked it when the doctor offered them an upside-down sheet of information as a method for being identified as LHL. Most participants agreed on the importance of presenting information layered and step-by-step for patients. GPs mentioned the importance of observing when people with LHL are saturated, or their attention span is narrowing. People with LHL suggested bringing a relative or friend to the consultation. Another helpful suggestion by people with LHL was encouraging patients or their proxies to take notes or record the consultation.

*It would be very nice to use those [decision aids] in a layered approach. Moreover, that you can be guided in the consultation by the patient, the questions he/she asks or the comments he/she makes that trigger you to think 'oh perhaps more information is needed' or that you notice, also from the patient's body language 'now it's enough'. [E2]*

**Prioritising patient context and an iterative SDM process.** Experts and GPs stated how important they regard the patient context. They discussed the importance of "step 0" to examine what patients already know about their disease and its medical options. It serves to find common solid ground for a shared understanding of the patient's problem at the consultation's beginning and sensitise the patient to the option talk information. The diagnosis or the problem posed by the patient should be clarified before options for intervention can be discussed.

*I will start by asking, 'What do you know about it yourself'? Do you already know something about it? Explain what you know about it. Because then I know what the patient's level of knowledge is at that moment and what words they are using, I can connect with that. At that level, I can continue. [G2]*

Several experts said a "step 0" would set the appropriate conditions for SDM. If not done already, this is the step in which the doctor has shared information on the diagnosis and subsequently may receive information from the patient's context and beliefs. People base their choices individually on information (cognitive) and/ or intuition (emotional). A "step 0" also informs on what form of decision-making process the patient prefers. What is necessary for him or her to decide? Maybe the presence of a relative is needed? Is there a need to distribute the decision-making process over multiple consultations?

*I think with patients with LHL, we need to emphasise even more that it's about doing it together. We are going to do this together, and we are going to help you. So, a more supportive role of the healthcare provider and not like throwing it over the fence. [E2]*

Some experts and GPs reported that doctors need a proactive attitude towards patients with LHL. Doctors should take a more active role in communicating understandably, and the focus on step 0 should be stronger. The consensus of

experts was that in step 0, patients should be given more space to express themselves rather than that the doctor immediately starts with the medical aspects.

*I think as a doctor, you just enter the conversation differently and maybe just be more active yourself, be more helpful, contrast the primary considerations more clearly, summarise, reflect instead of sending the whole thing across and then saying, 'the goal now is for you to make a choice'. I think that is always the goal of SDM, but the doctor needs to be even more focused on that in people with LHL. [E3]*

Some experts questioned the structure of SDM steps and suggested putting more value on the iterative approach with regard to the option and preference step to the extent of an almost reversed ordering. So, to start with the preference talk on what generally matters to the patient and tailor the options and their level of detail of risks and harms to what seems to fit what a patient considers a vital option.

*So, the thought that comes to me is that you guys want to do it nicely in the order of the model, first the information, then the preferences, but you might have to reverse it. [E1]*

One GP explained how to use the information from exploring the patient's reason for the encounter to tailor the presentation of options. It was emphasised that alternating between asking about the patient's preferences and explaining the options, and then checking whether these align with the patient's preferences would be beneficial.

*With someone with LHL, you use the information you already used when exploring the reason for the encounter, presenting the option, and testing ´Is that correct´? Then, you preselect a choice that you think fits best. You explain that one most clearly. And you don´t explain any other alternatives until you notice that there are questions or uncertainties about them. [G2]*

Most GPs preferred not to exclude options to prevent steering the patient based on withholding information. More emphasis on what is essential to the patient early in the decision-making process does not have to imply that the doctor excludes specific options.

*I think that's a very scary one. We know that, at least in the second line, clinicians can be very directive. You really do see options not being mentioned; I am very wary of that. [E11]*

Instead of deciding as a doctor that some options would not suit the patient, they can be investigated together. One GP indicated that all the options except the most "exotic" should be mentioned, but to skim over the options and move quickly towards the option that seemed to suit the patient best based on the context.

GPs apparently have the decisional power to give direction by explaining some options better than others, which is reinforced by the patients' attitude that "the doctor knows best". Some GPs take this decisional power and adopt a more paternalistic attitude. They assumed that such 'welcomed paternalism' holds the benefit of protecting the patient from information overload. Other GPs provide complete information on all the options and a thorough explanation for patients with LHL. Several GPs emphasised the importance of the iterative process of SDM as a dialogue between doctor and patient. People with LHL wished that doctors communicate better with each other to be better informed about the patient. They wished for more attention and consideration of their context and to be taken seriously.

*Well, I think […] that some options are indeed not mentioned because it is perceived that it might be too much to explain or too complicated. However, it doesn't have to be that way. I think it is also very much about how and how*

*much is explained, with what words, and whether it is supported with pictures. But that it […] is essential that you connect with the patient. […] What is important for that person for that patient? […] that it becomes a conversation rather than listing the options and then choosing. I don't think that's the way it works. But that it becomes a conversation […] and that you then go and discover together what is the most appropriate for [the patient] currently. [G8]*

**Critical views on the feasibility of option talk and RC in daily practice.** Patients with LHL emphasised how important it is that the doctor does not decide for them and should present all options to them. People with LHL welcomed it if the doctor was relaxed and reassuring during the explanation. They also discussed the importance of their GP as someone who knows the patient well. In the case of decision-making in secondary care, they would also prefer a consult with their GP about the options. Due to stronger doctor-patient relationships and knowledge about the patient's context, they would also welcome some advice on the best option from their GP. Most participants with LHL wished for tailored and practical personalised information in the consultation with their doctor.

*Well, first of all, a doctor cannot make decisions about your body. A doctor shouldn't, anyway. You are the boss of your own body. Look, I think like this: a doctor can say, 'I have two things for you: you can proceed with medication, but you can also proceed with an injection.' Look, I anyway don't think a doctor should decide for you. [P4]*

GPs agreed that images could be supportive as it is essential how options are explained and whether they are supported with images of body parts, such as ears, or with diseases, such as a herniated disc. One GP also mentioned that a good explanation, combined with images, could increase health literacy skills. A requirement for this was that there should be enough time to provide a thorough explanation. Several GPs said they got their images from Google©. Another participant stated that memorised images can also be used by drawing them.

*Sometimes, you use nice illustrations or icons you have in mind, but you don't always need to have them available. You can also draw them on paper yourself. [G2]*

The experts intensively debated about using numerical information in RC with LHL patients. Some experts stated that numerical information is explicitly needed to make an informed decision, whilst other experts said that there is other, maybe more critical information that patients use to base their decision on, like their religious beliefs, their past (treatment) experience, their surroundings, their emotional state and so on. A frequency or percentage cannot visualise all of these factors.

*So, the moment you know more about this patient, it becomes easy to make the medical information more meaningful for the patient. Making it concrete: 'What does this mean for the patient's life? [E11]*

Also, patients with LHL emphasised how important it is for the doctor to consider their context when discussing options. They stated that option talk is more important to them than all details on RC. Some people with LHL did not want to know all details on benefits and harms since it fans fear in them due to previously experienced side effects.

*You know, when doctors then say, 'You can get a blood clot'... Then you're already scared, zone out, stop listening, and don't want that operation anymore. [P4]*

However, patients with LHL also stated that the wish for details on harms and benefits depends on the person and that some people in their target group would want to know the details on harms and benefits. However, they experienced that RC does not often take place in practice. Also, for RC communication, they would prefer it if the GP tailored it to them and their context.

*There are further risks. [In how many] percentage [of people] can surgery happen? [...] Yes, if that [is] ten percent, then people can say, yes, I will go for surgery, but if that is 50 percent, then people will say, 'no.' [P3]*

**Preferences for specific RC strategies.** Explaining abstract concepts such as risk reduction is experienced as problematic by professionals. To explain risk, some GPs indicated that they make use of contextualising. In that case, they explain the risk metaphorically, or they use a comparison with a daily risk, such as '*the chance of dying during the operation is as low as getting hit by a car*'. Although some experts opted for contextualisation, others doubted the usefulness of this format. An argument against contextualisation was the risk of downplaying patients' fears. Their concerns can feel belittled. Another argument is that it can confuse when a risk is compared with a rare event.

*I sometimes compare it to the bus in the street where our practice is. It is a very narrow street and with a huge, huge bus. And then when I want to explain how rare something is, that's really rare, just like someone would once come under the bus in front of our practice. Ooh, Ooh, Ooh, ooh. Whether that estimate is entirely accurate, I doubt it. Or winning the state lottery, for example. Comparison to the lottery. That does help. [G3]*

*It is also problematic to make such a comparison because you then bring in other associations. So, when you say, 'Vaccinating is very safe because the chances of being struck by lightning are much higher´, what you then are doing is downplaying people's concerns. 'It is all nonsense'. [E1]*

People with LHL regarded contextualisation as not useful at all. They related it to the concept of uncertainty, that in every action, there are risks attached and that it would not add any useful information. They rather would not want to be aware of every single risk. The examples of RC strategies shown to the participants confirmed that some participants were confused that suddenly, it was spoken about a non-medical event such as falling from a bicycle. They did not understand the link between the metaphor and their risk of a disease or side effects.

*There are risks in everything. If I go to the ATM, there could be a madman walking in with a knife...yes. I could also fall off the pavement and break my neck or something; then it's also done. [...] I live by the day, so nobody has promised us a tomorrow. So yeah, I'm also like: Do I want to know everything? No. The doctor can't give me a guarantee either, so then I prefer him to keep his mouth shut. [P5]*

If numerical risks are to be communicated, the experts recommend using natural frequencies, e.g., 1 out of 100 people like you undergoing surgery get an infection due to the operation. For visualising risk, they recommend icon arrays. These were understandable for all patients, including patients with LHL. Some experts were involved in research on the understandability of RC, which was also aimed at patients with LHL.

*What works well are the icon arrays. All participants understand that, to our surprise. [E2]*

One GP said that despite years of experience, he is still trying to figure out how to deal with the abstract concept of risk in patients with LHL. In his experience, icon arrays cannot be regarded as the solution as these are too complicated.

*No, you know the diagrams you have in the example with the thousand boxes, or the hundred boxes or whatever. Even that is often very difficult for people who cannot handle such an abstract concept of risk. I don't know how to do it after 40 years as a doctor. [G3]*

Most patients with LHL regarded verbal RC only as too superficial since it led to ambiguity in interpreting what a "small" or "high risk" means. They preferred visual RC. They also preferred visual RC in combination with numerical RC over numerical risk information only. Icon arrays were seen to be more understandable than bar graphs.

*Pictures [visual RC] are much more useful. We are really picture people. [P2]*

**Critical views on the usability of PtDAs for people with LHL.** Experts and GPs were exposed to three PtDA formats. One was a multi-page web-based tool, another a so-called option grid that showed a table with options and outcomes on one A4 paper and another that was developed with patients with LHL and showed illustrations for the options on the first page and subsequently some details for each option but almost no RC. The experts mentioned that most patients with LHL need help using a PtDA by themselves before the consultation and that coaching by a healthcare professional is needed. The professionals mentioned that the PtDAs in the Netherlands are too complex for patients with LHL since they contain too much text.

*Many decision aids have been created that are very complicated and have a lot of text. I think they are complicated for all patients, let alone people with LHL. [G6]*

One GP wondered whether the option grids are suitable for patients with LHL.

*I actually don't use decision aids very often either, but sometimes I do use option grids. For instance, the one from Thuisarts on anticonception, this one I always like. However, I don't know whether it suits this target group. [G7]*

The PtDA developed together with patients with LHL was seen as a good tool as you can point to the illustrations and explain the steps. PtDAs usually include a choice table in the form of a matrix. Such a matrix is rather designed from the perspective of the doctor. It is a good overview for the healthcare provider but found to be too difficult for patients with LHL since it is much text in cells. The digital web-based format is helpful for doctors and some patients, but a group of patients still finds it hard to navigate online. Websites are often very information-rich. Therefore, the PtDA "Consultkaart in Beeld" was rated as the best PtDA to use for patients with LHL. According to some experts, using this format with illustrations for option talk was thought to support patients in becoming more talkative, asking questions, and understanding the options better.

*So, this [consultation card] works for patients with LHL who lack those school-based skills of being able to read a table, and that includes graphs and bar charts and that stuff, that just doesn't work well. What we see in our research is that the moment the healthcare provider uses visuals in a consultation, patients who normally participate less in the conversation start participating with more attention because it is presented in a way that is understandable to them. [E2]*

People with LHL saw PtDAs as a potentially helpful tool to open the conversation. They preferred to use PtDAs together with the doctor. They doubted that they would read them alone at home. They were also regarded as too difficult. However, if they had to read a PtDA at home, they would prefer to discuss it with the doctor again. They mentioned that the practice assistant could also support that process instead of the doctor.

*You often go home with things [information], and you don't know what to do with them [it]. [...] the important thing is that they [healthcare provider] take a moment for the low-literate person to explain what we take home on paper. Moreover, that is why those visuals are so important. [P1]*

People with LHL discussed that difficult words should be avoided, and the text should be a manageable amount. If a sentence starts with a difficult word and much text is used, this likely stops them from reading the information. People with LHL also emphasised how important the layout is and that the letter type and size of written information material must be adequate. Patients with LHL also repeatedly mentioned that computer skills must be considered, as some people have problems if the information is only digitally accessible.

## Discussion

### Main findings

This study shows the complexity of option talk and RC for LHL patients in practice and the influence of patient preferences. No single strategy can fix all challenges for patients with LHL. Nevertheless, essential views, common themes and misconceptions of experts, GPs, and people with LHL could be identified.

All participants emphasised the importance of thoroughly considering the patient context in the SDM process. Given the difficulty of identifying individuals with LHL in practice, the aim should always be to make conversations understandable for everyone. People with LHL emphasised the importance of considering each person's individuality and for doctors to take them seriously and understand their context and preferences. Adding a 'step 0' was regarded as necessary for SDM, in which the patient's level of knowledge, context and general goals are assessed to shape the option talk iteratively. Experts recommended clarifying patients' understanding of their disease and exploring the beliefs, preferences, and needs of patients with LHL. The dilemma of providing complete information during option talk versus purposefully withholding options to avoid information overload was intensely discussed. People with LHL explicitly stated they wanted to know all options but not necessarily detailed risk information, as it can cause fear. However, they highlighted the importance of tailoring to individual preferences since some patients with LHL explicitly seek detailed and quantitative risk information, whereas others prefer general information. Overall, people with LHL preferred to discuss treatment options and benefits and harms with their GP in a layered step-by-step manner, supported by visualisations such as icon arrays. Contextualisation, e.g., using metaphors, was considered confusing.

Comparison with other studies shows similar results, suggesting a modification to the SDM model by adding a 'step 0' to explicitly assess the patient's level of knowledge on, e.g., the diagnosis to better facilitate SDM [32]. Previous research has revealed the usefulness of RC strategies, such as icon arrays for people with LHL [20], aligning with our findings. In the SDM literature, RC is typically considered a sub-step of option talk. However, our findings suggest that they should be viewed as distinct steps, as patients wish to be informed about all treatment options, but preferences for quantitative RC vary.

In the Netherlands, several PtDAs for people with LHL have been developed using the same format "consultkaart in beeld". In this format, it is decided to focus on option talk without RC or only verbal RC. Since verbal RC alone can lead to ambiguity in interpretation and potentially biased decision-making [16], it is questionable if this format adequately informs patients. On the one hand, a simple format might be helpful for all patients (universal precaution approach) [33]. On the other hand, as our study suggests that some patients with LHL want detailed information, we probably need PtDAs tailored to the individual patient's information needs. There seems to be no one-size-fits-all strategy, and patient preferences for RC should be considered. However, the preference for an RC strategy and its effectiveness might differ [34]. More research is needed to explore the effectiveness of formats such as the "consultkaart in beeld".

PtDAs can support conversations when used appropriately by doctors, but they must be well-developed with the end-users, easy to read, and comprehensible [19]. Many online PtDAs contain extensive information and are difficult to navigate [28]. This could be challenging for people with LHL and those with lower digital skills [35,36]. The goal is to inform patients well based on their level of knowledge and by considering their context to facilitate a balanced decision. However, some strategies may be even more critical for people with LHL. Further research is required to explore the role of patient preferences in RC.

### Strengths and limitations

This is one of the few qualitative studies in RC research that show the perspectives of key stakeholders. The expert FGs were unbalanced regarding gender, which reflects the gender imbalance in the Dutch SDM research field. Six of the eight GPs are associated with the DCGP's working group on family medicine and international care; therefore, their experience could be linked to a specific patient type. The group of people with LHL was diverse, with a well-balanced gender ratio.

The member check after the FG with experts and GPs allowed participants to clarify or add information. Face-to-face FGs with individuals with LHL were held at an expertise centre for health literacy with a moderator experienced in working with this group. This ensured actively engaging people with LHL to explore their experiences and perspectives. Our focus groups used a single PtDA for knee arthritis, which may limit the generalizability of our findings to other conditions. While exploring PtDAs from different diseases could provide broader insights, our goal was not to examine disease-specific SDM outcomes but to offer recommendations on effective communication strategies, particularly for option talk and risk communication. Broader systemic factors, such as healthcare organization, time constraints, and external barriers were not the focus of our study. While these are challenges in implementing SDM, addressing them was beyond our scope. Given the specific needs of our target population, these systemic barriers could play a more significant role in real-world SDM processes, requiring further exploration in future research.

## Practical implications

It could be considered to integrate the exploration of the patient's level of knowledge and their context (step 0) more structurally into the process of SDM. This could help tailor option talk and RC to make numbers more meaningful to patients. Supporting option talk with illustrations seems to be a promising way to enhance understanding. We suggest using a stepwise, layered approach to provide information in option talk and RC. Patients' preferences for option talk and RC should be considered. Overall, the challenge of RC in patients with LHL needs to be investigated further. Natural frequencies and icon arrays seem currently to be the most promising RC strategies. However, preferences for RC and understandability and effectiveness of those strategies should be explored in representative groups of patients with LHL. The importance of the patient context needs to be further explored. More qualitative research is needed in RC, especially.

## Conclusions

Option talk and RC in practice continue to be challenging for doctors and patients with LHL. Key strategies identified in our study for people with LHL are likely relevant for all patients. Several strategies for option talk and RC have been identified: a) thoroughly consider the patient context in a more structured way in the SDM model (step 0); b) consider patient preferences for option talk and RC; c) use illustrations for option talk; d) use icon arrays for RC in combination with natural frequencies; e) PtDAs need to be tailored to the needs of people with LHL, f) future research should focus on RC with people with LHL in representative groups to explore further and confirm findings.

## Supporting information

**S1 Appendix.  PtDA examples.**
(PDF)

**S2 Appendix.  Examples used in focus groups with people with LHL.**
(PDF)

## Acknowledgments

 The authors thank the participants for their valuable contribution, the Dutch Centre of Expertise on Health Disparities (PHAROS) for their input on this project, and the decision aid developers for granting access to the decision aids.

## Author contributions

**Conceptualization:** Romy Richter, Maarten Jonkmans, Juliette Linskens, Jany Rademakers, Trudy van der Weijden, Jesse Jansen.

**Data curation:** Romy Richter, Maarten Jonkmans, Juliette Linskens, Jany Rademakers, Trudy van der Weijden.

**Formal analysis:** Romy Richter, Maarten Jonkmans, Juliette Linskens, Trudy van der Weijden.

**Funding acquisition:** Trudy van der Weijden, Jesse Jansen.

**Methodology:** Romy Richter, Maarten Jonkmans, Juliette Linskens, Jany Rademakers, Trudy van der Weijden.

**Supervision:** Romy Richter, Esther Giroldi, Jany Rademakers, Trudy van der Weijden, Jesse Jansen.

**Validation:** Romy Richter, Esther Giroldi, Juliette Linskens, Trudy van der Weijden, Jesse Jansen.

**Visualization:** Romy Richter, Esther Giroldi, Maarten Jonkmans, Juliette Linskens, Trudy van der Weijden, Jesse Jansen.

**Writing – original draft:** Romy Richter, Esther Giroldi, Maarten Jonkmans, Juliette Linskens.

**Writing – review & editing:** Romy Richter, Esther Giroldi, Jany Rademakers, Trudy van der Weijden, Jesse Jansen.

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
