## [Decision Letter · Decision Letter 0]

24 Sep 2024

Dear Dr. Richter,

Thank you for submitting your manuscript to PLOS ONE. After careful consideration, we feel that it has merit but does not fully meet PLOS ONE’s publication criteria as it currently stands. Therefore, we invite you to submit a revised version of the manuscript that addresses the points raised during the review process.

Please accept our apologies for the delay in returning an editorial decision to you. The manuscript has now been evaluated by two reviewers, and their comments are available below.



We look forward to receiving your revised manuscript.

Kind regards,

Steve Zimmerman, PhD

Senior Editor, PLOS ONE

Journal Requirements:

Reviewers' comments:

Reviewer's Responses to Questions

**Comments to the Author**

1. Is the manuscript technically sound, and do the data support the conclusions?

Reviewer #1: Yes

Reviewer #2: Yes

2. Has the statistical analysis been performed appropriately and rigorously?

Reviewer #1: N/A

Reviewer #2: Yes

3. Have the authors made all data underlying the findings in their manuscript fully available?

Reviewer #1: Yes

Reviewer #2: Yes

4. Is the manuscript presented in an intelligible fashion and written in standard English?

Reviewer #1: Yes

Reviewer #2: Yes

Reviewer #1: The study presents a topical issue, and its research is important if we want to apply, in a generalized way, less paternalistic and more participatory health models, where patients can make their own decisions. On the other hand, the application of qualitative methods helps to elicit patients' preferences and, in many cases, can offer results closer to the study objective than quantitative ones.

However, the difficulty in the design and the tools to be used on the target population are key elements. In this regard, from my point of view, the paper should explain in more detail the following points:

a) The choice of knee arthritis PtDA as an example.

It is deduced from reading the paper that the knee arthritis PtDA was used only as an example. Would it have been possible to work with several tools from different diseases with more/less aggressive treatments to analyze the possible consequences of SDM in different circumstances?

Undoubtedly, SDM will be more or less complex depending on the disease and its treatments, and using a specific example in the Focus Groups can overestimate or undervalue some results.

It is also important to consider that the target population of this work may be affected by social inequalities that affect, above all, preventive health measures. Adding an example related to these measures could be an alternative for a more concise understanding of patient preferences. Do the authors believe that the discussion and results would have been very different in this case?

This topic could be incorporated into the Discussion section.

b) Table 2 should be revised. So, for example:

1. The column with the number of participants in each GF does not correspond to the number cited in the abstract, nor does the number of rows correspond to the number of participants cited in the table heading.

2. We don't know which 2 experts dropped out of the study. Did this abandonment change the number of participants, or was it already considered at the time of preparing the table?

3. The number of people participating in the focus groups is low in relation to the usual average. The reason for this choice should be justified.

4. Participants in the expert and patient groups were subdivided into two FGs, although there were 11 and 6 people, respectively (an acceptable number for a single group). Explain why.

The facilitators and barriers of the health system organization are not reflected in any case in the paper. Only, on one occasion, the possible lack of time of the professionals is mentioned. The literature has repeatedly cited the importance of these elements faced by professionals when implementing the SDM. In addition, the specific needs of the target population mean that these external variables, related to the healthcare context, may have more relevance than in a study with another target population.

This topic could be addressed in the Discussion section.

Reviewer #2: Dear editor Thank you very much for choosing me as a referee. The article with the title "Belililil" is a very important and popular topic in the world. Although the article is well written, it is recommended that the following points be considered:

The purpose is clearly stated and relevant. It addresses an important issue in healthcare, focusing on shared decision-making (SDM) and risk communication (RC) for people with limited health literacy (LHL).

1. The methodology is well-defined, using focus groups (FGs) with different stakeholders. However, the sample size for people with LHL (2x n=3) is quite small, which might limit the generalizability of the findings.

2. The use of both inductive and deductive content analysis is appropriate for qualitative research.

3. The results are comprehensive and highlight key points such as the importance of tailoring SDM to the patient’s context and the challenges of identifying patients with LHL.

4. The variation in views on the level of detail required for effective RC is well-documented, showing the complexity of the issue.

5. The conclusion effectively summarizes the findings and suggests that the strategies identified for people with LHL may be relevant to a broader patient population.

6. The call for further research is appropriate, given the small sample size and the exploratory nature of the study.

7. The article is generally clear and well-structured. However, some sentences are quite long and could be broken down for better readability.

8. For example, the sentence “FGs included a brief presentation defining SDM, option talk, and RC and introducing different RC strategies and decision aids for guided group discussions” could be split into two sentences for clarity.

9. There are minor grammatical errors, such as “patient´s context” which should be “patient’s context.”

10. Ensure consistent use of abbreviations. For instance, “RC” is introduced without explanation in the “Methods” section.

11. The flow of the article is logical, moving from purpose to methods, results, and conclusion.

12. Ensure that each section transitions smoothly to the next. For example, the transition from the “Methods” to the “Results” section could be improved by adding a brief summary sentence.

13. Ensure all statements are backed by appropriate references. If any claims or findings are based on previous research, cite them accordingly.

14. Ensure consistent formatting throughout the document, including headings, subheadings, and bullet points.

15. Use bullet points or numbered lists where appropriate to enhance readability, especially in the “Results” section.

**Do you want your identity to be public for this peer review?** For information about this choice, including consent withdrawal, please see our Privacy Policy

Reviewer #1: No

Reviewer #2: No

---

## [Author Response · Author response to Decision Letter 1]

6 Nov 2024

Dear Steve Zimmerman,

Thank you for giving us the opportunity to revise and resubmit our manuscript to PLOS ONE.

We have provided a point-by-point response to reviewers and have specified where these changes appear in the manuscript. We used Track Changes to mark any changes made in the manuscript and uploaded a manuscript version with tracked changes, a clean manuscript version and the rebuttal letter. Please see below for our responses to the reviewer comments.

In response to the editor’s comments, we ensured that our manuscript meets PLOS ONE's style requirements, including those for file naming and added more information concerning the Data Availability Statement.

Sincerely,

Romy Richter, on behalf of all authors 

Reviewers' comments:

Reviewer #1:

Reviewer comment:

The study presents a topical issue, and its research is important if we want to apply, in a generalized way, less paternalistic and more participatory health models, where patients can make their own decisions. On the other hand, the application of qualitative methods helps to elicit patients' preferences and, in many cases, can offer results closer to the study objective than quantitative ones.

However, the difficulty in the design and the tools to be used on the target population are key elements. In this regard, from my point of view, the paper should explain in more detail the following points:

a) The choice of knee arthritis PtDA as an example.

It is deduced from reading the paper that the knee arthritis PtDA was used only as an example. Would it have been possible to work with several tools from different diseases with more/less aggressive treatments to analyze the possible consequences of SDM in different circumstances?

Undoubtedly, SDM will be more or less complex depending on the disease and its treatments, and using a specific example in the Focus Groups can overestimate or undervalue some results.

It is also important to consider that the target population of this work may be affected by social inequalities that affect, above all, preventive health measures. Adding an example related to these measures could be an alternative for a more concise understanding of patient preferences. Do the authors believe that the discussion and results would have been very different in this case?

This topic could be incorporated into the Discussion section.

Response to comment:

Thank you for your constructive comment. The choice of the knee arthritis patient decision aid (PtDA) was based on a small sample of newly developed PtDAs by the Dutch Centre of Expertise on Health Disparities (Pharos). At the time of our study, there were only four available PtDAs specifically designed for individuals with limited health literacy, and the knee arthritis PtDA was the only one that incorporated risk communication – specifically verbal risk communication. The other PtDAs did not include risk communication, which made the knee arthritis example particularly suitable for our analysis. We agree that SDM will be more or less complex depending on the disease and its treatments. We therefore regard knee arthritis a good example, as it is a condition that is easily understood by patients, as many can relate to or imagine the experience of knee pain. This helped us engage participants effectively during focus group discussions.

We selected this PtDA not because of the specific disease, but because of its unique approach to presenting information on treatment options and their benefits and harms to patients with limited health literacy. Our focus was on the communication strategies, not on the condition itself, allowing us to generate general recommendations for improving shared decision making (SDM) processes. Moreover, our work builds on previous research, including a comprehensive systematic review of risk communication strategies for people with limited health literacy, and an analysis of Dutch PtDAs, which has informed our approach. Therefore, we are confident that our findings are not overly influenced by the specific choice of the knee arthritis PtDA, but rather reflect broader, applicable insights.

Incorporating more examples from different diseases, including those involving preventive health measures, might indeed have provided additional insights. However, we believe the communication strategies we analyzed are broadly applicable across different health contexts. Our goal was not to demonstrate the specific consequences of shared decision making (SDM) in particular diseases, but rather to offer insights and recommendations on effective communication strategies, especially for option talk and risk communication. We appreciate your suggestion and will consider it for future work. Additionally, we will incorporate your comment into the Discussion limitation section to acknowledge the potential influence of disease context on SDM outcomes.

We agree that social inequalities can significantly impact preventive health measures. However, we believe that the communication strategies we examined, particularly for option talk and risk communication, are broadly applicable across different contexts, including preventive health. While the specific example used may influence certain nuances in patient responses, we do not expect the overall recommendations on communication strategies to change significantly. Exploring preventive health measures in future research would enrich understanding of shared decision making in these populations. However, this represents a different scope and would be more suited for a separate project.

Added text in discussion – strengths and limitations, see page 25:

Our focus groups used a single PtDA for knee arthritis, which may limit the generalizability of our findings to other conditions. While exploring PtDAs from different diseases could provide broader insights, our goal was not to examine disease-specific SDM outcomes but to offer recommendations on effective communication strategies, particularly for option talk and risk communication.

Reviewer comment:

b) Table 2 should be revised. So, for example:

1. The column with the number of participants in each GF does not correspond to the number cited in the abstract, nor does the number of rows correspond to the number of participants cited in the table heading.

2. We don't know which 2 experts dropped out of the study. Did this abandonment change the number of participants, or was it already considered at the time of preparing the table?

3. The number of people participating in the focus groups is low in relation to the usual average. The reason for this choice should be justified.

4. Participants in the expert and patient groups were subdivided into two FGs, although there were 11 and 6 people, respectively (an acceptable number for a single group). Explain why.

Response to comment:

1. We have revised the sentence in the abstract and added a legend to Table 2 to ensure the number of participants in each focus group (FG) aligns with the numbers cited in the abstract and the table instructions.

Methods: Five focus groups (FGs) were conducted with a) experts on health literacy, SDM, or risk communication (RC) (two FGs, n=5 and n=6); b) experienced doctors (one FG, n=8), and c) people with LHL (two FGs, 2x n=3).

Table legend, see page 11: *Focus groups with experts and people with LHL were divided into two groups. Grey shading indicates these divisions

2. The two experts who dropped out were accounted for when preparing the table. Therefore, the number of participants listed in Table 2 reflects the final number after these dropouts.

3. We acknowledge that the number of participants in some of the focus groups is lower than the typical average. This is specifically the case for the focus group with people with limited health literacy. We purposefully decided to keep the size of these FGs small, based on recommendations from the Dutch Centre of Expertise on Health Disparities (PHAROS) with expertise in working with people with limited health literacy. They advised smaller group sizes to foster more open discussion and richer data collection, which is particularly important when working with this population. Due to logistical constraints and experts' availability, we conducted two focus groups for experts. This ensured each expert could participate meaningfully without time conflicts. We believe that the other focus groups have a reasonable group size as shown in methodology literature. Meaningful discussions were reached also in the focus groups with less participants. As shown in methodology literature smaller focus groups work better when sensitive topics are discussed or when the participants have considerable expertise in or experience of the topic (Doody 2012).

- Plummer P. Focus group methodology. Part 1: Design considerations. Int J Ther Rehabil. 2017 Jul 2;24(7):297–301. Available from: https://www.magonlinelibrary.com/doi/abs/10.12968/ijtr.2017.24.7.297

- Doody O, Slevin E, Taggart L. Focus group interviews in nursing research: Part 1. British Journal of Nursing. 2013 Jan 9;22(1):16–9. Available from: https://pubmed.ncbi.nlm.nih.gov/23299206/

- Morgan D. Focus Groups as Qualitative Research. Focus Groups as Qualitative Research. SAGE Publications, Inc.; 2012. Available from: /record/1997-97533-000

1. The decision to subdivide the expert and patient groups into two focus groups was deliberate. For the experts, this approach was also due to practical reasons and logistical constraints. Given the full agenda and the availability of the experts, finding a single time slot that could accommodate everyone proved challenging. By creating two groups, we ensured that each expert could fully participate in a setting that allowed for meaningful contributions, without the risk of time constraints or scheduling conflicts. Additionally, we followed best practices, which suggest that focus groups should ideally have no more than six participants to encourage dynamic discussions. This approach was also applied to the focus group with people with limited health literacy to ensure deeper engagement and to create a comfortable environment for all participants to share their views.

We hope these explanations address your concerns and provide clarity regarding the design choices made for the focus groups.

Reviewer comment:

The facilitators and barriers of the health system organization are not reflected in any case in the paper. Only, on one occasion, the possible lack of time of the professionals is mentioned. The literature has repeatedly cited the importance of these elements faced by professionals when implementing the SDM. In addition, the specific needs of the target population mean that these external variables, related to the healthcare context, may have more relevance than in a study with another target population.

This topic could be addressed in the Discussion section.

Response to comment:

Thank you for your comment. Our primary focus was on developing recommendations for communication strategies, specifically regarding option talk and risk communication, rather than on examining broader facilitators and barriers to shared decision making (SDM). While we acknowledge the importance of systemic factors such as healthcare organization and time constraints, these were beyond the scope of our study. Our goal was to provide insights on improving communication within SDM, assuming that space and time are available for such consultations. However, we recognize the potential relevance of these external factors, particularly for our target population, and will address this briefly in the Discussion section.

Text in discussion, see page 25:

Broader systemic factors, related to the healthcare organization, such as time constraints, were not the focus of our study. While these are well-known challenges in implementing SDM, addressing them was beyond our scope. Given the specific needs of our target population, these systemic barriers could play a more significant role in real-world SDM processes, requiring further exploration in future research.

Reviewer #2:

Reviewer comment:

Dear editor Thank you very much for choosing me as a referee. The article with the title "Belililil" is a very important and popular topic in the world. Although the article is well written, it is recommended that the following points be considered:

The purpose is clearly stated and relevant. It addresses an important issue in healthcare, focusing on shared decision-making (SDM) and risk communication (RC) for people with limited health literacy (LHL).

Response to comment:

Thank you for your encouraging comment.

Reviewer comment:

1. The methodology is well-defined, using focus groups (FGs) with different stakeholders. However, the sample size for people with LHL (2x n=3) is quite small, which might limit the generalizability of the findings.

Response to comment:

We acknowledge that the number of participants in some of the focus groups is lower than the typical average. This it's the case for the focus group with people with limited health literacy that decision was based on recommendations from the Dutch Centre of Expertise on Health Disparities (PHAROS) with expertise in working with people with limited health literacy. They advised smaller group sizes to foster more open discussion and richer data collection, which is particularly important when working with this population. However, we believe that the other focus groups have a reasonable group size as shown in methodology literature such:

- Plummer P. Focus group methodology. Part 1: Design considerations. Int J Ther Rehabil [Internet]. 2017 Jul 2 [cited 2020 Aug 11];24(7):297–301. Available from: https://www.magonlinelibrary.com/doi/abs/10.12968/ijtr.2017.24.7.297

- Doody O, Slevin E, Taggart L. Focus group interviews in nursing research: Part 1. British Journal of Nursing [Internet]. 2013 Jan 9 [cited 2020 Aug 11];22(1):16–9. Available from: https://pubmed.ncbi.nlm.nih.gov/23299206/

- Morgan D. Focus Groups as Qualitative Research [Internet]. Focus Groups as Qualitative Research. SAGE Publications, Inc.; 2012 [cited 2020 Aug 11]. Available from: /record/1997-97533-000

The decision to subdivide the expert and patient groups into two focus groups was deliberate. For the expert group, we followed best practices, which suggest that focus groups should ideally have no more than six participants to encourage dynamic discussions. This approach was also applied to the focus group with people with limited health literacy to ensure deeper engagement and to create a comfortable environment for all participants to share their views.

Reviewer comment:

2. The use of both inductive and deductive content analysis is appropriate for qualitative research.

Response to comment:

Thank you very much for your feedback.

Reviewer comment:

3. The results are comprehensive and highlight key points such as the importance of tailoring SDM to the patient’s context and the challenges of identifying patients with LHL.

Response to comment:

Thank you very much for your feedback.

Reviewer comment:

4. The variation in views on the level of detail required for effective RC is well-documented, showing the complexity of the issue.

Response to comment:

Thank you very much for your feedback.

Reviewer comment:

5. The conclusion effectively summarizes the findings and suggests that the strategies identified for people with LHL may be relevant to a broader patient population.

Response to comment:

Thank you very much for your feedback.

Reviewer comment:

6. The call for further research is appropriate, given the small sample size and the exploratory nature of the study.

Response to comment:

Thank you very much for your feedback.

Reviewer comment:

7. The article is generally clear and well-structured. However, some sentences are quite long and could be broken down for better readability.

8. For example, the sentence “FGs included a brief presentation defining SDM, option talk, and RC and introducing different RC strategies and decision aids for guided group discussions” could be split into two sentences for clarity.

Response to comment:

Thank you very much for your valuable feedback. We have carefully reviewed the manuscript again, making a

---

## [Decision Letter · Decision Letter 1]

7 Jan 2025

Dear Dr. Richter,

Thank you for submitting your manuscript to PLOS ONE. After careful consideration, we feel that it has merit but does not fully meet PLOS ONE’s publication criteria as it currently stands. Therefore, we invite you to submit a revised version of the manuscript that addresses the points raised during the review process.

**ACADEMIC EDITOR:**

For acceptance details about metodology, as the charateristis of the particicpants of the FG.On of the reviewers have accepted this version, based on the chages were made, but the other reviewer has suggested some new information that could improve the manuscript.More details about the methodology should be clariflied. 

We look forward to receiving your revised manuscript.

Kind regards,

Marília Jesus Batista de Brito Mota, Post-doc

Academic Editor

PLOS ONE

Journal Requirements:

Additional Editor Comments:

The authors answered all the questions and suggestions of the first reviewers. The manuscript provides very relevant information on a subject that should be further explored in the literature. However, some points of the manuscript could be better clarified, especially some details of the methodology, which will be important for the study's reproducibility and better understanding.

Reviewers' comments:

Reviewer's Responses to Questions

**Comments to the Author**

Reviewer #1: All comments have been addressed

Reviewer #3: All comments have been addressed

2. Is the manuscript technically sound, and do the data support the conclusions?

Reviewer #1: Yes

Reviewer #3: No

3. Has the statistical analysis been performed appropriately and rigorously?

Reviewer #1: Yes

Reviewer #3: N/A

4. Have the authors made all data underlying the findings in their manuscript fully available?

Reviewer #1: Yes

Reviewer #3: No

5. Is the manuscript presented in an intelligible fashion and written in standard English?

Reviewer #1: Yes

Reviewer #3: Yes

Reviewer #1: (No Response)

Reviewer #3: I congratulate the authors for improving the quality of the manuscript and all the responses given to the reviewers of the first round of review. However, I believe that some additional points deserve to be answered, as follows:

Title:

- Title too long, remove: a focus group study with experts, doctors, and people with limited health literacy.

- Include the type of methodology.

Abstract:

- Make the objective shorter and more direct.

- The number of participants is a result (what was done) and not a method (how it was designed to include and/or exclude, select and/or calculate the sample)

- The methodology needs to have more information about the type of study, data collection method, sample selection method, when and where data collection took place, and data analysis method.

- Conclusion must respond to the objective of the study.

Introduction:

- Too long may be more limited to the purpose of the work.

- The purpose of the abstract and introduction are not standardized.

Methods:

- Having the estimated number of eligible people and how many were selected could improve understanding of the study.

- It talks about the recruitment period, but not about when data collection took place.

- How many interviewers? Was there prior training?

Results:

- How many were recruited? How many accepted? How many participated?

- If the research was carried out through a focus group, that is, through observation of the phenomenon through a group, why were the results presented individually?

**Do you want your identity to be public for this peer review?** For information about this choice, including consent withdrawal, please see our Privacy Policy

Reviewer #1: No

Reviewer #3: No

---

## [Author Response · Author response to Decision Letter 2]

6 Jul 2025

Reviewer comment:

Title:

- Title too long, remove: a focus group study with experts, doctors, and people with limited health literacy.

- Include the type of methodology.

Response to comment

Thank you for your comment. We adjusted the title to: "Option talk and risk communication with people with limited health literacy: a qualitative focus group study with key stakeholders.”

Reviewer comment:

Abstract:

- Make the objective shorter and more direct.

Response to comment

Thank you for your comment. We adjusted the objective to: To explore stakeholders´ views on acceptable and feasible strategies for discussion of treatment options and risk communication with people with limited health literacy (LHL in the context of shared decision-making (SDM)).

Reviewer comment:

Abstract:

- The number of participants is a result (what was done) and not a method (how it was designed to include and/or exclude, select and/or calculate the sample)

- The methodology needs to have more information about the type of study, data collection method, sample selection method, when and where data collection took place, and data analysis method.

- Conclusion must respond to the objective of the study.

Response to comment

Thank you for your valuable feedback. We have revised the abstract to better distinguish between methodology and results, ensuring that the number of participants is appropriately presented as a result. Additionally, we have expanded the methodology section to specify the study type and sampling strategy. Given the abstract's length constraints and the need to avoid information overload, we have focused on these key aspects. The data collection method (focus groups) and analysis approach (inductive and deductive content analysis) were already included. Further details on sample selection criteria, data collection timeframe, and location are provided in the methods section of the manuscript. Lastly, we have refined the conclusion to more directly address the study’s objective.

New abstract:

Purpose: To explore stakeholders´ views on acceptable and feasible strategies for discussion of treatment options and risk communication with people with limited health literacy (LHL in the context of shared decision-making (SDM)).

Methods: This qualitative descriptive study used purposive sampling to conduct focus groups with stakeholders including experts in health literacy, SDM, or risk communication (RC); experienced General Practitioners (GPs); and individuals with LHL. Each session included a brief presentation defining SDM, option talk, and RC, followed by an introduction to various RC strategies and decision aids to facilitate discussion. Verbatim transcripts were analysed by two independent researchers using inductive and deductive content analysis.

Results: Five focus groups were conducted, involving experts (two FGs, n=5 and n=6), GPs (one FG, n=8) and people with LHL (two FGs, 2x n=3). Experts and GPs emphasised the need to tailor communication to the patient’s context and noted challenges in identifying patients with LHL. All participants highlighted the importance of using illustrations of treatment options (e.g. knee injections) to support the discussion of options. Views on the level of detail required for RC varied, with some GPs questioning whether RC was understandable to people with LHL. Most people with LHL preferred RC in natural frequencies and icon arrays but noted that RC can fan fear. GPs found contextualisation (e.g. comparing the probability of a treatment outcome with the probability of a car accident) a helpful strategy, but patients found it confusing. Decision aids were seen as supportive for RC. Overall, people with LHL preferred their doctor to discuss options face-face with them, using a layered step-by-step manner, adding details on RC as preferred by patients.

Conclusion: SDM for people with LHL benefits from a tailored, layered approach with visual aids. These strategies are potentially useful for all patients, but further research is needed to confirm this.

Introduction:

- Too long may be more limited to the purpose of the work.

- The purpose of the abstract and introduction are not standardized.

Response to comment

Thank you for your feedback. We have shortened the introduction and checked for alignment with the purpose of the study. The adjusted introduction can be found in the main document.

Abstract:

To explore stakeholders´ views on acceptable and feasible strategies for discussion of treatment options and risk communication with people with limited health literacy (LHL in the context of shared decision-making (SDM)).

Introduction:

To gain an understanding of acceptable and feasible strategies to support SDM in practice, we aim to explore views and preferences regarding the discussion of treatment options (option talk) and RC among (a) experts in SDM, health literacy and RC, (b) experienced GPs, and (c) people with LHL.

Methods:

- Having the estimated number of eligible people and how many were selected could improve understanding of the study.

- It talks about the recruitment period, but not about when data collection took place.

- How many interviewers? Was there prior training?

Response to comment

Thank you for your feedback. We have added information on the data collection period in the methods section and provided further details on the interviewers under the "FG Interviews" subsection. Additionally, we have clarified the number of interviewers involved and whether prior training was conducted. Details on participant selection and recruitment were also added under Participant Selection and FG interviews (please see details under reviewer comment “results”)

Manuscript text

FG interviews

Due to the COVID-19 pandemic, FGs with experts and GPs took place online via the video platform Zoom©, all but two of the GPS who were approached participated in the study. The FGs with experts and GPs were conducted in January and February 2022. The FG with people with LHL took place at PHAROS in May 2022. The FGs were conducted in Dutch and lasted one and a half hours, with a break between the two main topics. FGs were audio and/or video recorded, and field notes were taken during the FGs with experts and GPs. We started with the two expert FGs, followed by the FG with the GPs and ended with the two FGs with people with LHL. A different interviewer was assigned to each participant group. The focus groups with individuals with LHL were conducted by an experienced interviewer from PHAROS. Focus groups with experts were led by co-author JR, a professor in health literacy, while those with GPs were conducted by co-author TvdW, a professor in shared decision-making. Both co-authors have extensive experience in conducting focus groups.

Results:

- 1) How many were recruited? How many accepted? How many participated?

- 2) If the research was carried out through a focus group, that is, through observation of the phenomenon through a group, why were the results presented individually?

Response to comment

1) Thank you for your feedback. We clarified those recruitment aspects. There were two dropout in the focus group with doctors due to personal reasons. Therefore, the number of participants presented in Table 2, plus the two dropouts in the doctor group, reflects the total number of individuals recruited.

Manuscript text:

Participant Selection

We used purposive sampling. Eleven experts and 30 GPs were contacted by email through the research team and their network from 01/10/2021 to 31/01/2022. To recruit GPs, names and contact details of potential participants were gathered in consultation with project advisors and with a working group on diversity and international health from the Dutch College of GPs (DCGP). All the experts and ten GPs responded positively to participating. Six people with LHL were recruited by The Dutch Centre of Expertise on Health Disparities (PHAROS) between February and May 2022. All participants had to be proficient in the Dutch language. Inclusion criteria can be found in Table 1.

FG interviews

Due to the COVID-19 pandemic, FGs with experts and GPs took place online via the video platform Zoom©, all but two of the GPs who were approached participated in the study.

2) Regarding the presentation of results, while focus groups capture group discussions, our analysis also considered individual perspectives within those discussions to provide a more comprehensive and nuanced understanding of the data. This approach highlights diverse viewpoints while still respecting the group context.

Manuscript text:

Data analyses: Although data were collected through focus groups, the presentation of results also attends to individual perspectives expressed within the group discussions. This approach enables a more comprehensive and nuanced understanding of the data by highlighting diverse viewpoints while preserving the group context.

---

## [Decision Letter · Decision Letter 2]

29 Jul 2025

Option talk and risk communication with people with limited health literacy: a qualitative focus group study with key stakeholders

PONE-D-24-06379R2

Dear Dr. Richter,

We’re pleased to inform you that your manuscript has been judged scientifically suitable for publication and will be formally accepted for publication once it meets all outstanding technical requirements.

Kind regards,

Marília Jesus Batista de Brito Mota, Post-doc

Academic Editor

PLOS ONE

Additional Editor Comments (optional):

The manuscript can be accepted for publication after these two rounds of revision, the authors having adressed with the reviewers' requests and comments.

Reviewers' comments:

Reviewer's Responses to Questions

**Comments to the Author**

Reviewer #3: All comments have been addressed

2. Is the manuscript technically sound, and do the data support the conclusions?

Reviewer #3: Yes

3. Has the statistical analysis been performed appropriately and rigorously?

Reviewer #3: N/A

4. Have the authors made all data underlying the findings in their manuscript fully available?

Reviewer #3: Yes

5. Is the manuscript presented in an intelligible fashion and written in standard English?

Reviewer #3: Yes

Reviewer #3: I congratulate the authors on their efforts to improve the quality of the new version of the manuscript. Therefore, I consider it suitable for acceptance.

**Do you want your identity to be public for this peer review?** For information about this choice, including consent withdrawal, please see our Privacy Policy

Reviewer #3: **Yes: ** Manoelito Ferreira Silva Junior

---

## [Editor Report · Acceptance letter]

PONE-D-24-06379R2

PLOS ONE

Dear Dr. Richter,

I'm pleased to inform you that your manuscript has been deemed suitable for publication in PLOS ONE. Congratulations! Your manuscript is now being handed over to our production team.

Kind regards,

on behalf of

Professor Marília Jesus Batista de Brito Mota

Academic Editor

PLOS ONE